# CoNRec: Context-Discerning Negative Recommendation with LLMs

## Abstract

Understanding what users like is relatively straightforward; understanding what users dislike, however, remains a challenging and underexplored problem. Research into users' negative preferences has gained increasing importance in modern recommendation systems. Numerous platforms have introduced explicit negative feedback mechanisms and leverage such signals to refine their recommendation models. Beyond traditional business metrics, user experience-driven metrics, such as negative feedback rates, have become critical indicators for evaluating system performance. However, most existing approaches primarily use negative feedback as an auxiliary signal to enhance positive recommendations, paying little attention to directly modeling negative interests, which can be highly valuable in offline applications. Moreover, due to the inherent sparsity of negative feedback data, models often suffer from context understanding biases induced by positive feedback dominance. To address these challenges, we propose the first large language model (LLM) framework for negative feedback modeling with special designed context-discerning modules. We use hierarchical semantic ID Representation to replaces text-based item descriptions and introduce an item-level alignment task that enhances the LLM's understanding of the semantic context behind negative feedback. Furthermore, we design a Progressive Group Relative Policy Optimization (GRPO) training paradigm that enables the model to dynamically balance the positive and negative behavioral context utilization. Besides, our investigation further reveals a fundamental misalignment between the conventional next-negative-item prediction objective and users' true negative preferences, which is heavily influenced by the system's recommendation order. To mitigate this, we propose a novel reward function and evaluation metric grounded in multi-day future negative feedback and their collaborative signals. Extensive experiments on a real-world industry-scale dataset from Taobao demonstrate that our method achieves state-of-the-art performance. Our work offers meaningful insights not only for the emerging field of negative feedback modeling but also for the broader recommendation community.

## 1 Introduction

In today's recommendation systems, empowering users to express negative preferences has become increasingly prevalent. Major e-commerce platforms such as Taobao, Pinduoduo, TikTok Shop, as well as video platforms like YouTube and TikTok, have implemented user-facing dislike buttons, allowing users to indicate items they do not like and thereby helping the system reduce similar recommendations in the future. This shift is driven by a growing emphasis on user experience, which is now considered equally important as traditional efficiency metrics such as Clickthrough rate and Gross Merchandise Value (Konovalova, 2024; Wang et al., 2023). In particular, negative feedback rates have emerged as a critical indicator of user satisfaction (Christian & Utama, 2021).

Meanwhile, in the domain of positive recommendation, advances in generative models and large language models have enabled the development of end-to-end recommendation systems, effectively addressing cold-start issues for both users and items (Deng et al., 2025; Wang et al., 2024a). Nonetheless, even with models as large as 1.8B parameters, these approaches still face severe response-time challenges. As a result, it appears more practical to deploy LLMs in offline settings, where negative item filtering emerges as a promising application to reduce the negative feedback rate.

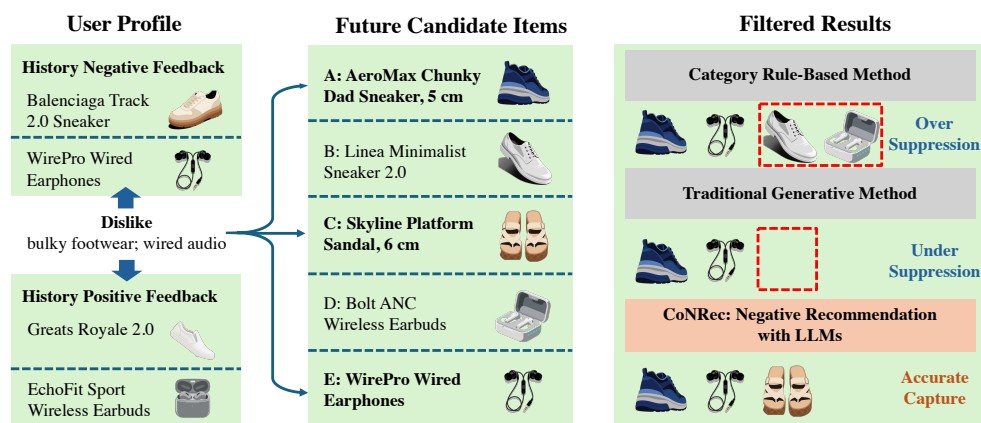

Figure 1: User Negative-Interest Modeling (icon generated by Doubao): For a user who dislikes bulky footwear and wired audio (A, C, E in bold), rule-based methods lead to over-suppression (red box represents wrong results) while traditional models perform poorly on cold-start items like bulky slippers (C), which never appear before. CoNRec effectively captures users' negative interests.

However, dedicated research on modeling users' negative interests remains limited. As shown in Fig. 1, Rule-based filtering strategies adopted by many platforms—such as completely blocking categories of items previously disliked by a user—often lead to excessive suppression. While the traditional generative method cannot deal well with the ID or embeddings it never sees and lacks of induction ability (Ding et al., 2024). To address these limitations, we propose a novel approach that leverages the world knowledge of LLMs to generate item list a user is likely to dislike, based on their historical behavior sequence—including both negatively rated items and positively interacted items (e.g., clicks, favorites, purchases). Our method replaces coarse rule-based filtering with a more nuanced, context-aware modeling of user dispreference. To the best of our knowledge, it is the first work to apply LLMs to the modeling of negative feedback in recommendation systems.

Directly applying current methods designed for positive recommendation to negative fields introduces several critical issues. First, reversing a positive recommendation model does not effectively create a negative recommendation model, because the absence of positive feedback usually indicates neutrality rather than explicit dislike (Cena et al., 2023). Second, while negative feedback sequences are far sparser than positive ones, they carry disproportionately high importance in shaping user experience. Existing models tend to be dominated by long sequences of positive interactions, causing them to overlook the critical signals in sparse negative feedback (Pan et al., 2023; Frolov & Oseledets, 2016). Lastly, standard fine-tuning objectives such as next-item prediction and evaluation metrics like hit rate are ill-suited for negative feedback tasks. In positive recommendation, the next interaction is often a strong indicator of user interest, reinforced by the system's feedback loop (Mansoury et al., 2020). However, in negative feedback scenarios, items previously disliked are typically filtered out permanently by current systems, eliminating any chance of re-appearance and subsequent user feedback. As a result, the next negatively interacted item is more influenced by the system's exposure mechanism than by the user's true negative preference, introducing significant noise into the training task and undermining the reliability of conventional evaluation metrics.

To address these challenges, this paper proposes the Context-Discerning Negative Recommendation with LLMs (CoNRec) framework. In order to enable the model to better understand negative-feedback contexts, CoNRec introduces an additional item-level fine-tuning task. This allows the model to focus more on potential negative factors of items without giving complex user historical behavior sequences. To prevent the model from being overly influenced by long sequences of positive feedback, CoNRec adopts a Progressive GRPO training paradigm that incrementally incorporates contextual information, ensuring that performance does not degrade compared to training solely with negative feedback. Furthermore, to address the inconsistency between a user's true interests and next negative feedback, CoNRec innovatively introduces the concepts of future and collaboration to transform the conventional next-item prediction into next-items. Based on this, we design a novel reward function and evaluation metrics tailored for the negative-feedback setting. CoNRec is particularly suited for the offline application for negative-feedback filtering, where the

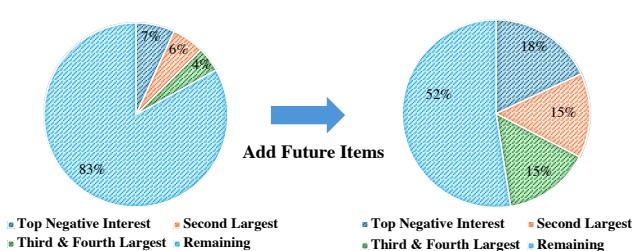 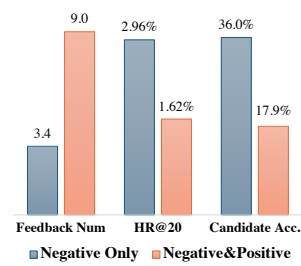

(a) Proportion of Main Negative Interest among Latest Feedback  (b) Performance Comparison

Figure 2: Illustration of motivational studies. (a) The next negative feedback item covers only 7% of the user's top negative interest (17% for Top-4), causing a lot of noise; while extending to a future 7-day horizon can raise Top-4 coverage to 48%. (b) Adding extra positive interactions (3× longer than negative) unexpectedly causes a large performance drop.

system screens target items by reconstructing to embedding representations and checks whether the maximum embedding similarity compared with the set generated by the model exceeds the threshold. In summary, the contributions of this paper are:

- To the best of our knowledge, CoNRec is the first large language model framework designed for negative item generation that models users' negative interests in the recommendation scenarios.
- To address current challenges in discerning context in negative recommendation, CoNRec introduces item-level alignment prior to the generation task and a progressive GRPO training paradigm. Our approach achieves state-of-the-art performance on real-world, industry-scale Taobao dataset.
- We break the limitation of predicting the next item in recommendation systems by extending to future items and collaborative items, and introduce new reward functions and evaluation metrics tailored to negative-feedback tasks, which also provide insights for general recommendation tasks.

## 2 MOTIVATIONAL STUDIES

As mentioned earlier, directly applying existing LLM methods in recommendation can lead to numerous issues. This section provides a quantitative analysis of the two critical problems, which also serve as key motivations for the following design of the CoNRec framework.

**Misalignment between User Negative Interest and Next Feedback Item.** The training data we collected in the negative feedback domain is biased. Under the existing negative feedback mechanism, items that have received multiple negative feedback from a user are likely to have been filtered out, making it difficult for the model to effectively capture the user's negative interests. As shown on the left of Fig. 2a, only 7% of the next negative feedback corresponds to the user's primary negative interest (approximated by categories), and even when considering the top four negative interests, the coverage rises to just 17%, introducing substantial noise into model fine-tuning. In contrast, when we expand the scope to include all negative feedback within the next 7 days, the coverage of the primary negative interest increases to 18%, and the top four interests reach 48%. This effectively mitigates the bias in negative feedback data and informs the design of CoNRec.

**Performance Drop with Extra Context.** Furthermore, we were surprised to find that adding a user's historical positive feedback sequences actually leads to a significant drop in model performance, which contradicts original intention of providing additional information to aid training. Due to the sparsity of negative feedback data, the length of positive feedback sequences is generally at least five times that of negative feedback sequences. This causes the model's attention to be disproportionately focused on the positive sequences, while the negative feedback sequences, which should be emphasized, are largely ignored. As shown in Fig. 2b, when using the LC-Rec framework (Zheng et al., 2024), incorporating both positive and negative feedback results in a 45% decrease in HR@20 compared to using only negative feedback sequences, and a 50% drop in candidate set accuracy in a simulated online environment. This indicates existing models cannot reasonably handle the relationship between negative and positive feedback sequences. Therefore, we aim to design a model that at least avoids performance degradation when additional information is introduced.

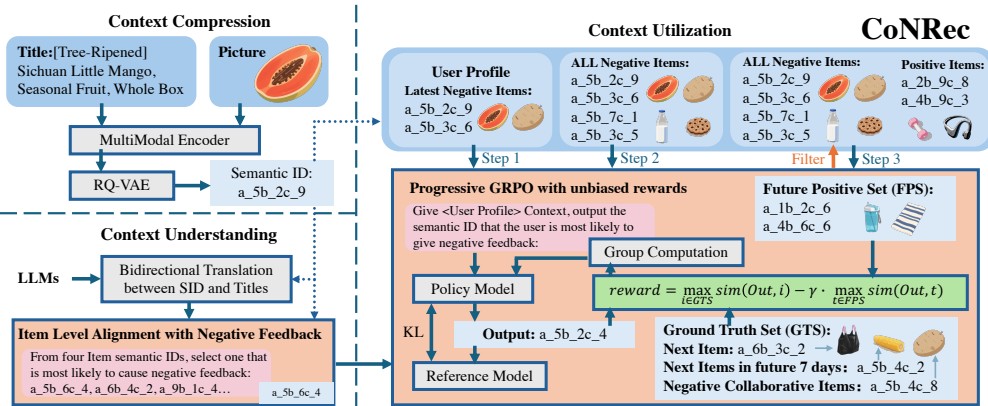

Figure 3: Overview of CoNRec framework. CoNRec first compresses the item information into semantic IDs, which are used in both Context Understanding stage and Context Utilization stage. Context Understanding includes the LoRA finetuning of traditional bidirectional translation, as well as proposed item-level alignment. Then, the model is post-trained using GRPO, where we progressively increase the complexity of the context during training, with a novel reward design that utilizes future negative, positive feedback and collaborative items to create an unbiased environment.

## 3 CoNREC

In this section, we formally define the task scenario of Negative Recommendation. Then we introduce our proposed framework, CoNRec, elaborating on its architecture and the mechanisms it employs to discern the complex contexts inherent in negative recommendation.

**Problem Formulation.** Let $\mathcal{U}$ denote the set of users and $\mathcal{I}$ the set of items. For each user $u \in \mathcal{U}$, we denote the sequence of items with which $u$ has interacted negatively as $\mathcal{N}_u = \{i_1, i_2, \ldots, i_m\}$ and the sequence of items with which $u$ has interacted positively as $\mathcal{P}_u = \{j_1, j_2, \ldots, j_n\}$, where all items are represented only by their semantic IDs. Given $\mathcal{N}_u$ and $\mathcal{P}_u$, the goal is to predict the set of items that user $u$ will most likely give negative feedback to in the future. Formally, we aim to learn a function $f : (\mathcal{N}_u, \mathcal{P}_u) \mapsto \hat{\mathcal{N}}_u^K$, where $\hat{\mathcal{N}}_u^K = \{\hat{i}_1, \hat{i}_2, \ldots, \hat{i}_K\}$ denotes the top-$K$ candidate items generated by the model (e.g., via beam search). We evaluate performance by comparing the generated set $\hat{\mathcal{N}}_u^K$ with the ground-truth negative feedback events observed in the future.

**Context Compression with Semantic ID.** In practical e-commerce, item titles are often redundant. To attract user attention, sellers tend to insert repeated fields or irrelevant information such as shipping details. This redundancy not only inflates the textual representation but also leads to excessively long contexts when fed into LLMs, exceeding their input limits. Moreover, models based solely on titles are restricted to discriminative tasks, since they lack a global notion of the item space, making them unsuitable for generative tasks (Singh et al., 2024). To represent items with compact and informative discrete codes, we introduce the concept of Semantic ID. As illustrated in the top-left of Fig. 3, a multimodal encoder integrates heterogeneous signals from the item title and image into unified embeddings, ensuring diverse content features are effectively captured (Radford et al., 2021).

These embeddings are passed through a Residual Quantized Variational Autoencoder (RQ-VAE) (Lee et al., 2022), which maps the multimodal input $x$ into a latent representation $X \in \mathbb{R}^d$ and discretizes it via multi-level residual quantization. At each level $d$, the residual $R^{(d)} = X - \sum_{i=1}^{d-1} Z^{(i)}$ is quantized by selecting the nearest codeword $Z^{(d)} \in \mathcal{C}^{(d)}$, yielding the final representation $\hat{X} = \sum_{d=1}^{D} Z^{(d)}$, from which the decoder reconstructs the embedding. The training objective combines reconstruction and quantization terms (van den Oord et al., 2018):

$$\mathcal{L} = \|X - \hat{X}\|_2^2 + \lambda \sum_{d=1}^{D} (\|R^{(d)} - \text{sg}[Z^{(d)}]\|_2^2 + \|\text{sg}[R^{(d)}] - Z^{(d)}\|_2^2) \tag{1}$$

where $\text{sg}[\cdot]$ denotes the stop-gradient operator, $\lambda$ is a balancing weight for the quantization loss. This design yields a hierarchical structure analogous to a category hierarchy in the form of Semantic IDs, which are going to be used in later Context Understanding and Context Utilization Stage.

**Context Understanding with Item Level Alignment.** CoNRec builds upon a general-purpose large language model (LLM) and initially employs a bidirectional translation task between Semantic IDs (SIDs) and item titles, a commonly adopted approach in methods leveraging SIDs, to help the LLM associate discrete SIDs with their corresponding textual meanings. However, we argue that this translation task alone does not adequately adapt to the negative feedback scenario, as the factors that make an item undesirable are not simply the inverse of those that make it attractive; much of the intervening space is neutral. To address this gap, we design an intermediate item-level alignment task using a LoRA-based supervised fine-tuning objective (Hu et al., 2021), which modifies LLM weight matrices $W$ by introducing low-rank matrices $A$ and $B$ such that the adapted weight becomes

$$W' = W + \Delta W = W + BA, \tag{2}$$

where $A \in \mathbb{R}^{r \times d}$ and $B \in \mathbb{R}^{d \times r}$ with rank $r \ll d$, enabling efficient adaptation with minimal additional parameters. Item Level Alignment focuses solely on item semantics without incorporating user profiles of historical behavior sequences. This approach reduces context length, making the training fast and lightweight. Concretely, we prompt the model as follows:

**[Prompt]** A user has negative feedback item A by clicking 'not interested', while simultaneously purchasing three other items B, C, and D. All items are represented by their Semantic IDs. Given four candidate Semantic IDs, determine which one most likely corresponds to the negative item A.

Through this task, the model is encouraged to contrast positive and negative signals, thereby learning to capture potential negative attributes of items at the semantic level.

**Context Utilization with Progressive GRPO and Unbiased Rewards.** After the Context Understanding stage, the model has acquired a certain level of awareness regarding the semantics of negative feedback. However, it still cannot fully address two key issues highlighted in motivational studies: the inconsistency between a user's next negative feedback and their genuine negative interests, and the performance interference introduced by positive sequence. In fact, all preceding modules are designed to better support the Context Utilization stage, where we introduce the Progressive GRPO module with an unbiased reward function to resolve the aforementioned problems.

In GRPO (Shao et al., 2024), given a context $c$, the previous policy model $\pi_{\theta_{\text{old}}}$ produces a set of $G$ candidate outputs $\{y_i\}_{i=1}^G$, for which the corresponding rewards $\{r_i\}_{i=1}^G$ are computed by the reward function. The optimization objective of the updated policy model $\pi_\theta$ is then formulated as:

$$\mathcal{L}_{\text{GRPO}}(\theta) = \mathbb{E}_{c \sim \mathcal{D}, \{y_i\}_{i=1}^G \sim \pi_{\theta_{\text{old}}}(\cdot|c)} \left[ \frac{1}{G} \sum_{i=1}^G \frac{1}{|y_i|} \sum_{t=1}^{|y_i|} \left( \min\left( \frac{\pi_\theta(y_{i,t}|c, y_{i,<t})}{\pi_{\theta_{\text{old}}}(y_{i,t}|c, y_{i,<t})} A_{i,t}, \right. \right. \right.$$
$$\left. \left. \left. clip\left( \frac{\pi_\theta(y_{i,t}|c, y_{i,<t})}{\pi_{\theta_{\text{old}}}(y_{i,t}|c, y_{i,<t})}, 1-\epsilon, 1+\epsilon \right) A_{i,t} \right) - \beta \mathbb{D}_{\text{KL}}\left( \pi_\theta \| \pi_{\text{ref}} \right) \right) \right] \tag{3}$$

where the advantage of $y_{i,t}$ is derived by applying a normalization over the rewards at group level:

$$A_{i,t} = \frac{r_i - \text{mean}\left(\{r_i\}_{i=1}^G\right)}{\text{std}\left(\{r_i\}_{i=1}^G\right)} \tag{4}$$

Note that the loss function in GRPO includes a KL divergence term, which prevents the model from deviating excessively from the original policy. Inspired by the concept of curriculum learning—from simple to complex—we divide the GRPO training process into three stages, where the length of context provided gradually increases. Specifically, in the first stage, we train the model using only the user's negative feedback sequence from the past three days. Then we expand the input to include the entire negative feedback sequence. Finally, we incorporate both negative and positive feedback contexts. By adjusting the value of $\beta$, we ensure that the model continues to gain performance improvements. Moreover, the model from earlier stages is used to assist in data cleaning for later stages. For instance, if the generated items are found to be overly similar to those receiving future positive feedback, CoNRec applies data augmentation to such samples.

As shown on the right of Fig. 2a, when the ground truth is extended from the next negative feedback item to the next items within the following 7 days, the coverage of the user's top-4 negative interests reaches 48%, significantly reducing the inconsistency between the ground truth and the user's actual negative interests. In the GRPO reward computation, we further expand the set to include high-collaboration items from the user's actual negative feedback, obtained using the Swing algorithm

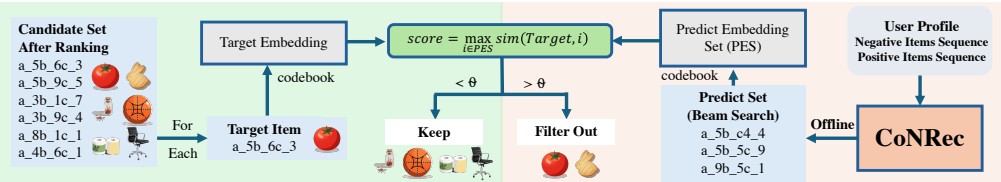

Figure 4: Illustration of CoNRec's offline industrial application. The target item from the ranking stage and the items predicted by CoNRec are reconstructed into embeddings via the stored codebook. A similarity score is then computed as the maximum embedding similarity between the target and predicted items. Items with scores exceeding the threshold are filtered out.

(Yang et al., 2020) from the negative feedback data:

$$\text{Swi}(i,j) = \sum_{u \in U(i) \cap U(j)} \sum_{v \in U(i) \cap U(j), v \neq u} \frac{1}{(|I(u)| + \alpha_1)^\theta (|I(v)| + \alpha_1)^\theta} \cdot \frac{Id(|U(i) \cap U(j)|)}{|I(u) \cap I(v)| + \alpha_2} \cdot \frac{1}{\sqrt{N_j}}$$

$$Id(x) = \begin{cases} 1, & x > 5 \\ 0, & x \le 5 \end{cases} \tag{5}$$

where $U(i)$ and $U(j)$ are the sets of users who interacted with items $i$ and $j$; $I(u)$ is the set of negative-feedback items for user $u$; $Id(x)$ is a threshold function to filter out low-collaboration user pairs; and $\alpha_1, \alpha_2, \theta$ are smoothing or weighting hyperparameters. With this expansion, the coverage of the user's top-1 negative interest reaches 25%, and top-4 reaches 64%, greatly reducing noise interference. As shown in bottom-right of Fig. 3, we first map all semantic IDs back to embedding representations via the codebook. The reward is then calculated using cosine similarity between the generated output and the Ground Truth Set, while penalizing similarity with the Future Positive Set:

$$reward = \max_{i \in GTS} sim(Out, i) - \gamma \cdot \max_{t \in FPS} sim(Out, t) \tag{6}$$

Through progressive input and improved reward function, CoNRec successfully avoids two major issues highlighted in motivational studies, achieving precise modeling of users' negative interests.

**Industrial Application.** One key advantage of CoNRec lies in its capability to generate results offline, allowing it to function as a filtering mechanism and thus avoiding excessive latency during online inference. As illustrated in Fig. 4, given a user's sequence of positive and negative interactions, CoNRec first generates a Predict Set via beam search. Subsequently, both the ranking-stage target item and the predicted items are reconstructed into embedding representations using the stored codebook. The maximum similarity between these embeddings is then computed and used as the score, and any item with a score exceeding a predefined threshold is discarded.

## 4 EXPERIMENTS

This section presents comprehensive experimental results, providing evidence that the challenges outlined in the motivational studies have been effectively addressed, leading to SOTA performance.

**Evaluation Metrics.** Performance is assessed by verifying whether the generated item set matches ground-truth negative feedback in future interactions. As motivational studies note, relying solely on the first future negative feedback introduces bias from existing negative feedback filtering mechanisms and recommendation order. To mitigate this, similar to the reward setting, we define ground truth as any negative feedback within the next seven days and propose FHR@20, counting a hit if any such item is predicted. Furthermore, for practical and stability reasons, platforms often adopt LLM-based models as incremental enhancements rather than replacements, mainly benefiting cold-start scenarios. Hence, full-sample evaluation is less meaningful in such scenario. We therefore introduce two additional FHR@20 metrics: LUF@20 for long-tail users (fewer than three negative-feedback instances; 20% of data) and LIF@20 for long-tail items (fewer than five instances; 12% of data). We also report Candidate Accuracy, where the model selects the true negative-feedback item from 20 candidates (one true and 19 distractors) (Kim et al., 2024), closely simulating online deployment. Traditional NDCG (Järvelin & Kekäläinen, 2002) is excluded, as it is unsuitable for negative-feedback tasks where ranking relevance is undefined and irrelevant to filtering task.

Table 1: Performance comparison on the Real-world Taobao Dataset. The best results are shown in **bold**, and the best baseline is underlined. HR@20 denotes the top-20 Hit Ratio against the next feedback. FHR@20 denotes the top-20 Hit Ratio against the user's feedback in the following 7 days. LUF@20 refers to FHR@20 measured on long-tail users with fewer than three historical negative feedbacks. LIF@20 refers to FHR@20 measured on long-tail items with fewer than five historical negative feedbacks. Candidate Acc. indicates the accuracy of a selection task simulating the online scenario where 20 items are presented and only one corresponds to the user's negative feedback.

| Model | HR@20 | FHR@20 | LUF@20 | LIF@20 | Candidate Acc. |
|---|---|---|---|---|---|
| Item ID based Generative Methods | | | | | |
| Caser | 0.0098 | 0.0128 | 0.0085 | 0.0135 | N/A |
| SASRec | 0.0180 | 0.0262 | 0.0169 | 0.0280 | N/A |
| BERT4Rec | 0.0186 | 0.0260 | 0.0173 | 0.0311 | N/A |
| Item Feature based Generative Methods | | | | | |
| FDSA | 0.0284 | 0.0374 | 0.0232 | 0.0362 | N/A |
| $S^3$-Rec | 0.0268 | 0.0329 | 0.0206 | 0.0382 | N/A |
| Semantic ID based Generative Methods | | | | | |
| P5-CID | 0.0262 | 0.0381 | 0.0220 | 0.0356 | N/A |
| TIGER | 0.0264 | 0.0388 | 0.0232 | 0.0360 | N/A |
| Item Title based LLM Methods | | | | | |
| TALLRec | N/A | N/A | N/A | N/A | 0.2686 |
| InstructRec | N/A | N/A | N/A | N/A | 0.3453 |
| Semantic ID based LLM Methods | | | | | |
| LC-Rec (Neg.&Pos.) | 0.0159 | 0.0381 | 0.0199 | 0.0351 | 0.1333 |
| LC-Rec (Neg. Only) | 0.0296 | 0.0385 | 0.0258 | 0.0397 | 0.2892 |
| **CoNRec** | **0.0330** | **0.0441** | **0.0297** | **0.0496** | **0.6950** |
| Improv. | +11.5% | +13.7% | +15.1% | +24.9% | +101.3% |

**Experiment Setup.** We utilize real-world, industrial-level data derived from user behavior logs on Taobao and discarding dislikes because of repetition and price factors to simply model interests. For Context Compression, we use a three-level codebook with 8192 dimensions at each level. For following SFT and Post-training, we use Qwen3-14B as a backbone. During the Bidirectional Translation stage, a corpus of 10 million pairs of commonly used semantic IDs and product titles is employed. For the Item-level Alignment stage, 4 million multiple-choice samples are constructed; in each sample, one item is selected as the correct option if it appears in the user's historical negative-feedback list more than three times, while three distractor options are randomly drawn from the user's historical purchase sequence. In the Progressive GRPO stage, we first perform a warm-up SFT using 200K samples, followed by 100K samples for post-training in each subsequent stage. To evaluate Candidate Accuracy, a 100K-sample test set is created where each user's actual negative-feedback item is the correct answer, and 19 distractors are randomly sampled from the user's daily exposure, forming a 20-choice task to assess the model's ability to capture negative preferences.

**Compared Methods.** We compare our approach against five categories of sequence recommendation methods: (1) generative models based on item IDs, including Caser (Tang & Wang, 2018), SASRec (Kang & McAuley, 2018), and BERT4Rec (Sun et al., 2019); (2) models that additionally incorporate item features such as text or images, represented by FDSA (Zhang et al., 2019) and $S^3$-Rec (Zhou et al., 2020); (3) generative methods leveraging semantic IDs, such as TIGER (Rajput et al., 2023) and P5-CID (Geng et al., 2023); (4) LLM-based methods utilizing product titles, including TALLRec (Bao et al., 2023) and InstructRec (Zhang et al., 2023), which are evaluated only on discriminative tasks due to their lack of item-space awareness; and (5) the latest LLM approach integrating semantic IDs, LC-Rec (Zheng et al., 2024).

**Quantitative Results.** As shown in Table 1, on Taobao's real-world user negative-feedback dataset, generative methods incorporating both item IDs and item features outperform those using only item IDs, particularly on cold-start metrics LUF@20 and LIF@20, due to the richer semantic information

Table 2: Ablation study evaluating the effectiveness of each module in CoNRec. The table is incremental, with each row adding one additional module on top of the previous configuration.

| CoNRec | HR@20 | FHR@20 | LUF@20 | LIF@20 | Candidate Acc. |
|---|---|---|---|---|---|
| baseline | 0.0286 | 0.0364 | 0.0246 | 0.0382 | 0.2764 |
| + Item Level Alignment | 0.0294 | 0.0382 | 0.0262 | 0.0410 | 0.5088 |
| + Progressive-Context GRPO | 0.0308 | 0.0393 | 0.0262 | 0.0428 | 0.5476 |
| + Negative Future Rewards | 0.0330 | 0.0434 | 0.0268 | 0.0452 | 0.6532 |
| **+ Positive Future Rewards** | **0.0330** | **0.0441** | **0.0297** | **0.0496** | **0.6950** |

provided by item features. Among generative approaches, encoder–decoder models based on semantic IDs achieve the best performance, demonstrating the strong information compression capability of semantic IDs, which makes them highly suitable for recommendation tasks. Under settings where item sequences are constructed using semantic IDs, LLM-based models leverage reasoning ability to consistently surpass encoder–decoder methods on cold-start metrics. Our proposed CoNRec model combines the strengths of semantic IDs and LLMs while avoiding the limitations observed in LC-Rec under negative-feedback scenarios, achieving 11.5% improvement on HR@20 and 13.7% on FHR@20—indicating that FHR is a more generalizable metric. In the most advantageous cold-start scenario, CoNRec yields 15.1% improvement for long-tail users and 24.9% for long-tail items. Finally, in Candidate Accuracy—simulating an online deployment setting—CoNRec demonstrates exceptional performance, doubling the results of the previous best model, InstructRec, highlighting its strong potential for possible online application.

**Ablation Studies.** As shown in Table 2, we incrementally add the modules proposed in CoNRec to the baseline LLM model trained with bidirectional translation to validate their effectiveness. Incorporating Item-level Alignment yields a 4–8% improvement on generative metrics while substantially boosts discriminative task accuracy. Introducing progressive context input does not underperform compared to using only negative sequences and even delivers a slight gain. Expanding the ground truth from the first future negative feedback to a seven-day window as well as high-collaboration items further improves HR@20 series metrics by 5–10%. Finally, applying a penalty based on future positive-feedback similarity leads to the best overall performance with the full CoNRec model.

**Forgetting Rate for Item-Level Alignment.** From the perspective of transfer learning, the higher the alignment between the target task and the intermediate task, the lower the forgetting rate of the intermediate task (Lin et al., 2024). The Item-Level Alignment task identifies items that users strongly dislike and strongly like, with high confidence and minimal noise. Therefore, if the model forgets a large portion of the intermediate task during target

Table 3: Forgetting Rate at Different Tasks.

| Task | Acc. | Forget Rate |
|---|---|---|
| Item Level Alignment | 0.755 | N/A |
| Traditional PNI SFT | 0.518 | 31.4% |
| Traditional GRPO | 0.540 | 28.5% |
| CoNRec | 0.652 | **13.6%** |

task training, we can infer that the target task contains considerable noise that interferes with the intermediate task. We compared the traditional predict-next-item SFT with the GRPO that calculates reward similarity based solely on the first negative feedback. As shown in Table 3, CoNRec exhibits the lowest forgetting rate, indicating our design can substantially reduce task noise.

**Reward Schemes Analysis.** The design of the reward is critical for GRPO. We explored several reward schemes based on an extended Ground Truth Set. For approaches using embedding similarity, we considered:(a) rewarding the similarity between the generated output and future negative feedback; (b) rewarding only high similarity, setting rewards below 0.6 to zero; (c) rewarding similarity with future negative feedback while simultaneously penalizing similarity with future positive feedback;

Table 4: Performance on Different Rewards.

| Scheme | FHR@20 | LUF@20 | LIF@20 |
|---|---|---|---|
| a | 0.0434 | 0.0268 | 0.0452 |
| b | 0.0421 | 0.0268 | 0.0450 |
| c | **0.0441** | **0.0297** | **0.0496** |
| d | 0.0438 | 0.0297 | 0.0496 |
| e | 0.0397 | 0.0264 | 0.0432 |

(d) truncating rewards for both future negative and positive feedback. For approaches using hit-based rewards, we tested: (e) rewarding 1 for hitting a level-3 semantic ID, 0.1 for level-2, and 0.01 for level-1. As shown in Table 4, reward truncation has minimal impact on the results, whereas incorporating penalties for future positive feedback leads to a notable improvement in the LUF metric. In scheme (e), due to the sparsity of rewards, both convergence speed and final performance are suboptimal. Based on overall performance, we selected scheme (c) as the preferred reward design.

## 5 RELATED WORK

**LLM-based Recommendation.** Recent advances in recommendation systems increasingly leverage large language models (LLMs) to improve item retrieval and ranking by incorporating richer contextual signals. Text-based methods rely on product descriptions, reviews, or metadata to capture semantic relationships between users and items, taking advantage of LLMs' ability to understand natural language and learn context-aware representations (Wang et al., 2024a; Bao et al., 2023; Dong et al., 2024). However, these approaches often suffer from high computational costs and sensitivity to noisy or sparse text, particularly in large-scale settings. An alternative line of work employs semantic IDs—compact dense vectors that encode item semantics using multimodal information and user interaction data (Zheng et al., 2024; Wang et al., 2024c). These representations enhance efficiency, scalability, and response speed in real-time recommendations while avoiding heavy reliance on textual data. Despite these advantages, most existing studies primarily address positive-feedback scenarios (Zhang et al., 2024). Extending LLM-based and semantic ID-based methods to negative-feedback modeling remains non-trivial due to the inherent sparsity, noisiness, and distinct characteristics of negative signals, which pose unique challenges for model training and evaluation.

**Negative-aware Recommendation.** Recent years have seen increasing interest in leveraging negative feedback to enhance the quality and robustness of recommender systems. Existing approaches incorporate signals such as dislikes, skips, low ratings, or survey responses in various ways (Yu et al., 2025). Some methods integrate explicit negative feedback into training objectives, introducing loss functions that penalize ranking negatively interacted items too highly (Wang et al., 2023). Others refine feedback interpretation by using large language models (LLMs) to distinguish true negatives from mislabeled ones, improving label accuracy (Pei et al., 2024; Shimizu et al., 2025). Graph-based models have also adopted negative-feedback-aware message passing, where interaction polarity guides information propagation (Wang et al., 2024b). These strategies demonstrate that negative signals can improve user modeling and overall recommendation performance. However, most methods still treat negative feedback as auxiliary information for refining positive preference modeling, rather than directly capturing user dissatisfaction or aversion. Even when explicitly incorporated, negative signals often reweight training samples or adjust positive representations instead of addressing distinct drivers of dislikes. This leaves a critical gap: few approaches directly model negative user experiences as a primary objective to proactively prevent undesirable recommendations and improve long-term satisfaction.

## 6 CONCLUSION

In this work, we introduced CoNRec, a novel recommendation framework that integrates semantic IDs with large language models to address the limitations of existing methods in negative-feedback scenarios. Comprehensive experiments on large-scale industrial datasets demonstrate that CoNRec consistently improves both standard and cold-start offline metrics, achieving substantial gains over state-of-the-art baselines. Furthermore, the proposed reward functions and evaluation metrics mitigate noise by extending the prediction objective beyond the immediate next item to encompass future items and their collaborative counterparts, providing a more generalizable and practical framework applicable across diverse recommendation settings, including those involving negative feedback. In addition, CoNRec exhibits strong potential in enhancing candidate accuracy metrics designed to approximate online performance. Future work will focus on investigating the feasibility and effectiveness of deploying negative-feedback-aware models in real-world production environments, both offline and online.

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

CONTENTS

# A  RQ-VAE Implementation

This appendix provides the technical implementation details and algorithmic procedure for the Residual Quantized Variational Autoencoder (RQ-VAE) used for generating Semantic IDs, complementing the high-level overview in Section 3.

## A.1  RQ-VAE Architecture and Training

The RQ-VAE consists of three primary components: a multimodal encoder $E$, a residual quantizer $Q$, and a decoder $D$. The encoder $E$ maps the input data (e.g., multimodal item features) to a continuous latent representation $X = E(x)$. The core of the model is the residual quantization process, which iteratively approximates $X$ through a series of vector quantizations.

Given a hierarchy of $D$ codebooks $\{\mathcal{C}^{(1)}, \mathcal{C}^{(2)}, \ldots, \mathcal{C}^{(D)}\}$, each containing $K$ codewords, the quantization proceeds layer-by-layer. The quantization process is defined recursively. At each step $d$, the algorithm quantizes the residual from the previous step and updates the residual for the next iteration. This hierarchical approach allows the model to capture details at multiple levels of abstraction.

The training objective combines two loss components:

**Reconstruction Loss**: Ensures the quantized representation $\hat{X}$ accurately reconstructs the original input embedding.

**Quantization Loss**: Consists of two terms that (1) bring the codewords closer to the residual representations and (2) encourage the residuals to align with their corresponding codewords.

The stop-gradient operator ($\text{sg}[\cdot]$) is crucial for stable training, preventing the quantization loss from distorting the encoder's representations.

---

**Algorithm 1** RQ-VAE Forward Pass and Training

---

**Require:** Multimodal input $x$, encoder $E$, codebooks $\{\mathcal{C}^{(1)}, \ldots, \mathcal{C}^{(D)}\}$, decoder $D$
**Ensure:** Quantized representation $\hat{X}$, reconstruction $\hat{x}$, loss $\mathcal{L}$
1: **Step 1: Encode Input**
2: $X \leftarrow E(x)$ {Encode to continuous representation}
3: **Step 2: Hierarchical Quantization**
4: Initialize residual: $R^{(1)} \leftarrow X$
5: Initialize quantized representation: $\hat{X} \leftarrow \mathbf{0}$
6: Initialize semantic ID: $S \leftarrow [\,]$ {Empty list for codeword indices}
7: **for** $d = 1$ to $D$ **do**
8:     Find nearest codeword: $k^{(d)} \leftarrow \arg\min_k \|R^{(d)} - c_k^{(d)}\|_2$ where $c_k^{(d)} \in \mathcal{C}^{(d)}$
9:     $Z^{(d)} \leftarrow c_{k^{(d)}}^{(d)}$ {Select codeword}
10:    Append to semantic ID: $S.\text{append}(k^{(d)})$
11:    Update quantized representation: $\hat{X} \leftarrow \hat{X} + Z^{(d)}$
12:    Compute next residual: $R^{(d+1)} \leftarrow R^{(d)} - Z^{(d)}$
13: **end for**
14: **Step 3: Decode and Compute Loss**
15: $\hat{x} \leftarrow D(\hat{X})$ {Reconstruct input from quantized representation}
16: $\mathcal{L}_{\text{recon}} \leftarrow \|x - \hat{x}\|_2^2$ {Reconstruction loss}
17: $\mathcal{L}_{\text{quant}} \leftarrow 0$
18: $R^{(1)} \leftarrow X$ {Reset residual for loss calculation}
19: **for** $d = 1$ to $D$ **do**
20:    $\mathcal{L}_{\text{quant}} \leftarrow \mathcal{L}_{\text{quant}} + \|R^{(d)} - \text{sg}[Z^{(d)}]\|_2^2$ {Codebook loss}
21:    $\mathcal{L}_{\text{quant}} \leftarrow \mathcal{L}_{\text{quant}} + \|\text{sg}[R^{(d)}] - Z^{(d)}\|_2^2$ {Commitment loss}
22:    $R^{(d+1)} \leftarrow R^{(d)} - \text{sg}[Z^{(d)}]$ {Update residual}
23: **end for**
24: $\mathcal{L} \leftarrow \mathcal{L}_{\text{recon}} + \lambda \cdot \mathcal{L}_{\text{quant}}$ {Total loss}
25: **return** $\hat{X}, S, \hat{x}, \mathcal{L}$ {Return quantized representation, semantic ID, reconstruction, and loss}

---

## A.2 Implementation Notes

The algorithm generates Semantic ID $S$ as a sequence of codeword indices $[k^{(1)}, k^{(2)}, \ldots, k^{(D)}]$, forming a hierarchical discrete representation. During inference, only steps 1-2 are needed to generate the Semantic ID for an item. The hyperparameter $\lambda$ balances reconstruction fidelity and quantization regularity. Codebooks are updated using exponential moving averages during training, following standard vector quantization practices (van den Oord et al., 2018).

# B   More Experimental Details

## B.1   Data Cleaning

The user behavior log data from Taobao Mobile contains the reasons for users' negative feedback, which are as follows: "Not wanting to see this product", "Not wanting to see this category", "Have viewed/purchased", "Not wanting to see this store", "Uncomfortable with product images", "Suspected AI-generated images", "Low price to trick clicks", and "Suspected counterfeit goods".

Since CoNRec is designed to capture users' negative interests, we only retain negative feedback samples related to user interests, namely "Not wanting to see this product", "Not wanting to see this category", "Not wanting to see this store", and "Uncomfortable with product images". Other negative feedback caused by repeated recommendations or product quality issues is not used for CoNRec training. Filtering such negative feedback (which does not stem from negative interests) is the responsibility of other methods based on user fatigue or rule-based mechanisms.

## B.2   Baseline Models

Here, we briefly introduce the principles of various models that are used to compare with CoNRec in the experimental section of the main text.

- **Caser** (Tang & Wang, 2018) is a CNN-based approach that models user behaviors through the application of horizontal and vertical convolutional filters.
- **SASRec** (Kang & McAuley, 2018) is a unidirectional self-attentive sequential recommendation method that captures long-range dependencies in user behavior sequences using Transformer architectures.
- **BERT4Rec** (Sun et al., 2019) is a bidirectional Transformer model that leverages BERT-like pre-training to capture both forward and backward dependencies in user behavior sequences.
- **FDSA** (Zhang et al., 2019) is a hybrid method combining both items and item features with self-attentive layers to model sequential patterns in user-item interactions.
- **S$^3$-Rec** (Zhou et al., 2020) employs mutual information maximization for pre-training a self-supervised sequential recommendation model, capturing the associations between items and their attributes.
- **P5-CID** (Geng et al., 2023) structures a variety of recommendation tasks into a text-to-text framework and uses the T5 model to uniformly handle different tasks. The research team then investigates the development of item indexing mechanisms for sequential recommendation scenarios, such as sequential indexing and collaborative indexing. In our work, we adopt P5 with collaborative indexing as a reference model.
- **TIGER** (Rajput et al., 2023) employs a generative retrieval framework for sequential recommendation tasks, incorporating the semantic id to provide unique item recognition.
- **TALLRec** (Bao et al., 2023) is an efficient tuning framework that aligns LLMs with recommendation tasks via fine-tuning on recommendation data—addressing gaps from LLM-recommendation task mismatches and insufficient pre-training data.
- **InstructRec** (Zhang et al., 2023) is a instruction-tuned recommendation framework that aligns model behavior with natural language instructions for better controllability.
- **LC-Rec** (Zheng et al., 2024) is an LLM-based recommendation model that integrates language and collaborative semantics by using learning-based vector quantization for meaningful item indexing and specially designed tuning tasks to enhance collaborative semantic integration, enabling direct item generation from the entire set without relying on candidates.

## C    LLM BACKBONE SETTING

We compared the performance of CoNRec on TBStars007-13B, Qwen3-8B, and Qwen3-14B. Experiments show that in the task of capturing users' negative interests, the performance of the TBStars007-13B model and the Qwen3-14B model is roughly comparable; meanwhile, the Qwen3-14B model achieves a slight performance improvement compared to the smaller-parameter Qwen3-8B model. However, regardless of which LLM backbone is adopted, CoNRec consistently demonstrates stable performance enhancement, which reflects the generalization capability of the CoNRec model.

Table 5: FHR@20 on Different Backbones.

| Backbone | LC-Rec | CoNRec |
|---|---|---|
| TBStars007-13B | 0.0381 | **0.0439** |
| Qwen3-8B | 0.0362 | **0.0410** |
| Qwen3-14B | 0.0385 | **0.0441** |

## D    LIMITATION

Although CoNRec has achieved state-of-the-art (SOTA) performance across a range of offline metrics and demonstrates a particularly significant improvement effect for users with a small number of historical negative feedback, it performs poorly for users who have no historical negative feedback at all and only have historical positive feedback. For this specific user group, we are attempting to enhance the effectiveness of contrastive learning in the phase of Context Understanding with Item Level Alignment, as well as adjust the penalty settings for future positive feedback in the GRPO phase. Our aim is to enable the model to learn from historical positive feedback and summarize predictions of potential negative future feedback.

## E    THE USE OF LARGE LANGUAGE MODELS

No large language models (LLMs), including but not limited to ChatGPT, GPT-4, Claude, and Llama, were utilized in any stage of the research and writing process of this paper. All content presented in this work—encompassing the formulation of research questions, design of methodology, analysis of experimental data, drafting of the main text, compilation of references, and preparation of appendices—was independently conceived, developed, and written by the authors.