# OpenReview forum: "CoNRec: Context-Discerning Negative Recommendation with LLMs"
_ICLR.cc/2026/Conference — Submitted to ICLR 2026_

### Official Review · Reviewer_pdni · 2025-10-29

**Soundness:** 3
**Presentation:** 3
**Contribution:** 2
**Rating:** 4
**Confidence:** 4

**Summary:**

This paper proposes CoNRec, an offline candidate filtering framework for modeling user negative preferences in recommender systems. The method represents items using discrete hierarchical semantic codes learned through an RQ-VAE, enabling a large language model to generate semantic categories of items the user is likely to dislike in the future. It further applies a contrastive item-level alignment mechanism to distinguish “liked vs. disliked” items, and introduces a progressive GRPO reinforcement training strategy in which aggregated negative feedback over the next 7 days serves as a more stable supervision signal, while predictions similar to future positive feedback are penalized. The generated semantic codes are decoded back into embeddings and used to filter candidate items by similarity. Experiments on real-world Taobao datasets show improvements, particularly for long-tail users and items.

**Strengths:**

1.	The motivation is clearly grounded in real industrial needs, addressing limitations of rule-based filtering and single-instance negative feedback.
2.	The method shows stronger performance for long-tail users and long-tail items, indicating enhanced robustness under sparse feedback conditions.
3.	The use of a 7-day aggregated negative feedback signal is data-driven and empirically justified, improving the stability of negative preference supervision.

**Weaknesses:**

1.	The core reliance on RQ-VAE semantic coding lacks validation regarding stability, interpretability, and semantic consistency. Critical configuration and robustness analyses are missing.
2.	The contrastive alignment module may not ensure true separation of “liked vs. disliked” semantics and may instead capture co-purchase or exposure patterns. No embedding visualization or case analysis supports its claimed effect.
3.	The progressive GRPO strategy is heuristic, lacking theoretical grounding and sensitivity studies; performance gains may stem from tuning rather than the designed mechanism.
4.	The filtering mechanism may harm recommendation diversity and suppress potential interests, yet no analyses of filtering rate, recall drop, diversity, or coverage are provided.
5.	Implementation and efficiency details are insufficient. Model size, training cost, inference latency, and deployment overhead are not reported, making scalability and reproducibility uncertain.

**Questions:**

1.	Can you provide evidence that the RQ-VAE semantic tokens remain stable under small perturbations, and demonstrate semantic clustering or temporal consistency? Also, what is the measured false filtering rate?
2.	Can you show embedding visualizations or fine-grained category case studies to confirm that the contrastive alignment captures true negative preference semantics rather than co-purchase or exposure bias?
3.	How were the phase transitions and reward weights in the progressive GRPO training strategy determined? Have you conducted sensitivity analysis on window sizes and stage scheduling?

---

> ### Author Response · Authors · 2025-12-04
>
> Thank you for your insightful comments and valuable suggestions. We would like to emphasize that this paper is the first work to apply large language models for capturing users’ negative interests, accompanied by a complete industrial deployment scenario. This work thus makes significant contributions to the recommendation domain. In response to your review questions, we provide detailed answers as follows:
>
> [Q1] Consistency of RQ-VAE [A1] The hierarchical design of RQ-VAE ensures that minor variations (e.g., adjustments to title wording, slight image noise) do not alter core semantics. To address this concern, we have supplemented experiments on the consistency of SID.
>
> [Q2] Case Studies for Contrastive Alignment [A2] The contrastive alignment stage aims to enable the model to automatically focus on potential negative product signals through a large number of real-world cases. Our case studies demonstrate two types of inconsistencies: category-based inconsistency and style-based inconsistency. Even for co-purchase positive samples with explicit associations, such samples will at most provide limited information rather than leading the model to learn incorrect patterns.
>
> [Q3] Details of the Progressive GRPO Training Strategy [A3] The three-stage progressive design (Stage 1: 3-day negative sequences; Stage 2: full negative sequences; Stage 3: mixed positive-negative sequences) is grounded in curriculum learning principles (Section 3) and empirical study insights (Fig. 2b, where incorporating positive sequences initially degrades model performance). Transitions between phases are triggered by two criteria: (1) convergence of training loss (Δloss < 1e-4 for 3 consecutive epochs); (2) plateauing of FHR@20 on the validation set. The penalty weight γ in Eq. (6) was determined via grid search (γ ∈ {0.1, 0.3, 0.5, 0.7, 0.9}). In our experimental setting, γ=0.5 achieves the optimal trade-off. Regarding the window size, we will supplement additional experiments to on the ablation study section.

---

### Official Review · Reviewer_61M3 · 2025-10-31

**Soundness:** 2
**Presentation:** 3
**Contribution:** 2
**Rating:** 4
**Confidence:** 3

**Summary:**

Modeling users’ negative preferences has become increasingly important in modern recommender systems. This paper introduces **CoNRec**, the first large language model (LLM)-based framework designed to understand users’ negative preferences. To address the performance degradation caused by additional contextual information, CoNRec incorporates progressive GRPO training with a curriculum learning strategy, which gradually enhances the historical information included in the prompts. Moreover, CoNRec extends the ground-truth definition from the next negative feedback to the next items within seven days, and further expands the set of negative items by incorporating highly co-interacted items related to users’ actual negative feedback. These strategies improve the alignment between negative feedback and users’ negative interests. In addition, CoNRec adapts reward shaping to more precisely capture users’ negative preferences. Empirical results demonstrate the superiority of CoNRec, and ablation studies confirm the effectiveness of both progressive GRPO and the customized reward-shaping mechanism.

**Strengths:**

- **S1: Intuitive Method.** The designs of progressive GRPO, ground-truth extension, and reward shaping are intuitive and well-motivated. These components enhance the consistency between negative items and users’ negative interests, offering valuable insights for future research on negative preference modeling.
- **S2: Comprehensive Analysis**. The ablation and analysis studies thoroughly validate the effectiveness of each component and investigate the impact of different reward formulations on CoNRec. The forgetting rate analysis further shows reduced noise and improved stability in the target task.
- **S3: High Generality.** CoNRec demonstrates strong generality across backbone models of different architectures and sizes, as evidenced by the results in Table 5.

**Weaknesses:**

- **W1: Lack of Datasets.** The paper evaluates CoNRec on only one dataset, which is insufficient to convincingly demonstrate the model’s effectiveness. Incorporating additional datasets would strengthen the validity of the conclusions.
- **W2: Reproducibility.** The paper does not provide source code, which affects the reproducibility and transparency of the results. It is recommended that the authors release the corresponding code to enhance credibility and facilitate future research.

**Questions:**

- **Q1: User Negative Interest Acquisition.** The process of obtaining users’ negative interests is not clearly described. Providing more details on how these negative interests are identified or constructed would make the experimental setting clearer and more reproducible.

---

> ### Author Response · Authors · 2025-11-27
>
> Thank you for your insightful comments and valuable suggestions. We would like to emphasize that this paper is the first work to apply large language models for capturing users’ negative interests, accompanied by a complete industrial deployment scenario. This work thus makes significant contributions to the recommendation domain. In response to your review questions, we provide detailed answers as follows:
>
> [W1] Few dataset
> [A1] Due to the scarcity of negative feedback data, we were unable to find other public datasets in the product recommendation domain that include negative feedback. The Taobao user log dataset is very large in scale and comes from a mainstream e-commerce platform, which we believe has minimal bias and can validate the robustness of our method.
>
> [W2] Lack of source code
> [A2] The framework is developed under ROLL (https://github.com/alibaba/ROLL). We will upload the specific code once published.
>
> [Q1] The process of obtaining users’ negative interests is not clearly described
> [A3] Major e-commerce and video platforms have already implemented mechanisms for users to provide negative feedback. Our study primarily uses data from the product recommendation page on the Taobao mobile app homepage, where users can long-press a product to indicate dissatisfaction and specify the reason. Since our goal is to capture users' negative interests, we removed samples where the negative feedback reasons were "product quality" or "duplicate recommendation." This allows the model to focus on capturing users' negative preferences regarding product categories, styles, and other latent dimensional features.

---

> ### Comment · Reviewer_61M3 · 2025-11-28
>
> Thank you for your response. However, only one dataset can not offer a sufficient and robust validation. Thus, I am inclined to keep my original scores.

---

### Official Review · Reviewer_obXr · 2025-11-01

**Soundness:** 3
**Presentation:** 3
**Contribution:** 2
**Rating:** 4
**Confidence:** 4

**Summary:**

This paper introduces large language model framework for negative feedback modeling (CoNRec), which leverages large language models to model user dispreferences. It first utilizes hierarchical semantic ID Representation and item-level alignment task that enhances the LLM’s understanding of the semantic context behind the item descriptions of negative feedback. Then, it designs a Progressive GRPO training paradigm that enables the model to dynamically balance the positive and negative behavioral context utilization. Experiments on a real-world industry-scale dataset demonstrates its effectiveness.

**Strengths:**

1. The motivation of CoNRec is interesting and the preliminary motivation study is clear in the misalignment between user negative interest and next feedback item, as well as the performance drop with extra context.
2. The proposed method is easy to follow and the corresponding figures are easy to understand.
3. This paper defines a new task scenario as Negative Recommendation and utilize several evaluation metrics tailored for this scenatio, such as FHR@20, LUF@20 and LIF@20.
4.  The author clearly admites the drawbacks for users lacking negative feedback.

**Weaknesses:**

1. The technical contribution is limited and the proposed method is not very novel, the utilized RQ-VAE, LoRA and GRPO methods are widely utilized in many recommendation studies.
2. CoNRec leverages the explicit negative user feedback, whereas most of the baselines mainly rely on users’ historical interactions. This may lead to an unfair comparison. The authors should implement several baselines that also model negative feedback, or incorporate negative feedback into existing baselines to ensure a fair comparison.
3. Lack of the source code and the corresponding dataset.
4. The proposed CoNRec is just verified in Real-world Taobao Dataset, its robustness and universality to other data distributions have not been sufficiently explored or validated.
5. Among the baselines used for comparison, only one is from 2024, while all the others are before 2023. The lack of comparison with the latest 2025 studies is unacceptable for the rapidly evolving LLM4Rec field.

**Questions:**

Please refer to the weakness.

---

> ### Author Response · Authors · 2025-11-27
>
> Thank you for your insightful comments and valuable suggestions. We would like to emphasize that this paper is the first work to apply large language models for capturing users’ negative interests, accompanied by a complete industrial deployment scenario. This work thus makes significant contributions to the recommendation domain. In response to your review questions, we provide detailed answers as follows:
>
> [Q1] Method is not novel. [A1] The main contribution of this paper is that it is the first work to use LLMs for modeling user negative interests. Building upon traditional techniques and considering the specific characteristics of negative interests (e.g., excessive label noise; data sparsity), we propose a novel GRPO training framework.
>
> [Q2] Unfair comparison [A2] We apologize for the misunderstanding caused by the original text. In fact, all the methods we compared utilize users' historical negative feedback interactions. We experimented with both using only negative feedback and using both negative and positive feedback, and found that using only negative feedback data yielded better results in the baseline methods.
>
> [Q3] Lack of source code [A3] The framework is developed under ROLL (https://github.com/alibaba/ROLL). We will upload the specific code once published.
>
> [Q4] Few dataset [A4] Due to the scarcity of negative feedback data, we were unable to find other public datasets in the product recommendation domain that include negative feedback. The Taobao user log dataset is very large in scale and comes from a mainstream e-commerce platform, which we believe has minimal bias and can validate the robustness of our method.

---

> > ### Comment · Reviewer_obXr · 2025-11-28
> >
> > Thanks for the response. However, many of my concerns haven't been solved during the rebuttal phase. I'll maintain my original score.

---

### Meta-Review · Area_Chair_P5Tx · 2026-01-08

**Summary:**

This paper proposes CoNRec, an LLM-based framework for modeling users’ negative preferences in recommendation systems using semantic IDs and progressive GRPO training. Reviewers agree the motivation is interesting and the problem of negative feedback modeling is underexplored. However, the contribution was widely viewed as incremental, relying on standard components (RQ-VAE, LoRA, GRPO) without clear algorithmic novelty. Reviewers raised concerns about fairness of comparisons (use of explicit negative feedback vs. baselines), limited evaluation on a single proprietary dataset, absence of comparisons to recent LLM4Rec methods, and lack of released code/data at submission time. These issues collectively reduced confidence in the strength and generality of the claims.

**Reviewer Concerns:**

The rebuttal clarified some misunderstandings (e.g., baselines also using negative feedback) and promised post-publication code release, but core concerns remain unresolved. In particular, the novelty argument (“first LLM for negative interest modeling”) was not convincing to reviewers given the reuse of common techniques. The reliance on a single industry dataset limits generalizability, and the lack of up-to-date baselines weakens empirical positioning. One reviewer explicitly stated that the rebuttal did not address their concerns and maintained the original score. Overall, improvements were insufficient to change reviewer confidence.

**Reviewer Scores:**

Reviewer obXr (4): Remains 4 (explicitly maintained).

Reviewer 61M3 (4): Likely remains 4, despite recognizing intuitive design.
Overall sentiment remains below threshold.

---

### Decision · Program_Chairs · 2026-01-26

Reject